# UQE: A Query Engine for Unstructured Databases

**Hanjun Dai**[†]♪, **Bethany Yixin Wang**[‡]♪, **Xingchen Wan**[‡], **Bo Dai**[†]¶, **Sherry Yang**[†],
**Azade Nova**[†], **Pengcheng Yin**[†], **Phitchaya Mangpo Phothilimthana**[†]*,
**Charles Sutton**[†], **Dale Schuurmans**[†§]

[†] Google DeepMind [‡] Google Cloud [§] University of Alberta ¶ Georgia Institute of Technology

## Abstract

Analytics on structured data is a mature field with many successful methods. However, most real world data exists in unstructured form, such as images and conversations. We investigate the potential of Large Language Models (LLMs) to enable *unstructured* data analytics. In particular, we propose a new Universal Query Engine (UQE) that directly interrogates and draws insights from unstructured data collections. This engine accepts queries in a Universal Query Language (UQL), a dialect of SQL that provides full natural language flexibility in specifying conditions and operators. The new engine leverages the ability of LLMs to conduct analysis of unstructured data, while also allowing us to exploit advances in sampling and optimization techniques to achieve efficient and accurate query execution. In addition, we borrow techniques from classical compiler theory to better orchestrate the workflow between sampling methods and foundation model calls. We demonstrate the efficiency of UQE on data analytics across different modalities, including images, dialogs and reviews, across a range of useful query types, including conditional aggregation, semantic retrieval and abstraction aggregation.

## 1 Introduction

Data analysis [13] is essential for making well founded decisions and enabling businesses and society to function more effectively. Relational databases [12, 32] and the Structured Query Language (SQL) [7] have delivered huge successes in *structured data* management and analysis. Typically, such data is collected and organized in a pre-defined schema [14], where the data properties and relationships have been pre-specified, and downstream analysis is restricted to this schema.

In most real-world applications, however, data exists in unstructured formats, such as images, documents and audio recordings. Without preprocessing such data into structured forms, traditional SQL engines can only support limited queries. Preprocessing, including document entity retreival [45] and form understanding [42], also require training on downstream tasks given a predefined taxonomy. This naturally motivates the question we consider in this paper:

*How can one perform unstructured data analysis in a **flexible** and **efficient** way?*

In the literature, full-text search engines [20] support scalable regexp-matching search on unstructured data, but this becomes infeasible for more complex semantic reasoning queries. Retrieval-Augmented Generation (RAG) [39, 24, 18] allows question answering on a subset of related data, but is not directly applicable to generic analytical tasks with aggregation and semantic queries that spans over an entire large database. Recent advances in Large Language Models (LLMs) [4, 2] unlock the ability to perform flexible question answering, especially with recent long-context models [33]. However, setting aside the cost per query, data analytics can still be challenging for LLMs without fine-tuning [25] or few-shot demonstrations [9], even given structured tables.

---

*work done while Phitchaya Mangpo Phothilimthana was at Google DeepMind.

♪ Correspondence to `hadai@google.com` and `wyixin@google.com`.

38th Conference on Neural Information Processing Systems (NeurIPS 2024).

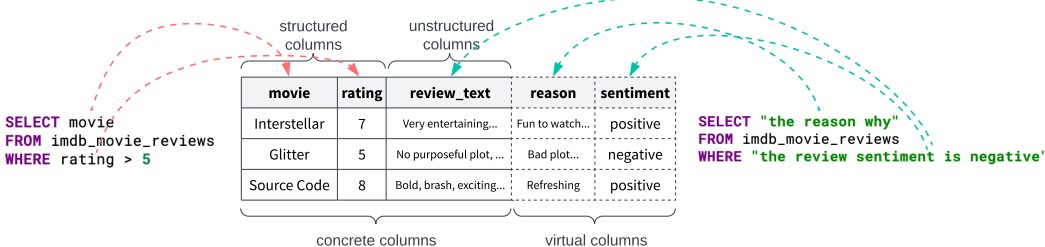

Figure 1: Illustration of unstructured data analysis defined in Section 2.

Recently, a promising line of work considered marrying LLMs with programming frameworks [38], where logical or arithmetic operations are offloaded to program interpreters [17]. A most relevant example for analytics and table understanding tasks is Cheng et al. [11], which augments classical SQL semantics with LLMs as user-defined functions (UDF). While promising, the execution of such SQL programs that embed LLM calls still requires sweeping over the entire database, which is too costly for large collections of unstructured content. To overcome this barrier, we leverage the synergy between LLMs and programmatic execution to define an Unstructured Query Language (UQL) that augments SQL for flexible semantic queries, with a focus on improving scalability and efficiency.

A key observation is that the efficiency of classical SQL engines relies on (1) indexing structures that avoid the need to scan the entire database, and (2) a compilation system that determines the best execution order for operations. Based on these ideas, we propose the Unstructured Query Engine (UQE), which refines and extends this design principle to unstructured data analytics. To achieve similar effect to indexing, UQE casts the problem as learning to search or sample, seeking to avoid a full database scan with statistically sound methods. Additionally, a compilation system is developed that determines the best execution order and operator combination for different clauses in a UQL query, with the goal of minimizing LLM calls while preserving query semantics.

As part of this project, we have created four new benchmark datasets, with both text and image modalities, along with three common analytic tasks. Compared to baseline methods, such as long-context LLMs and embedding based retrieval, UQE achieves significant improvements in terms of the accuracy and cost reduction on these benchmarks.

## 2 Problem

Before defining the problem we are solving, we first establish the terminology and notation we will use throughout the paper. A concrete illustration of the following terms is given in Figure 1.

- **Table / database**: We define a table $\mathcal{T} = \{\mathcal{T}_i\}_{i=1}^{N}$ as an unordered set of $|\mathcal{T}| = N$ rows, where each row $\mathcal{T}_i = [\mathcal{T}_{i,1}, \mathcal{T}_{i,2}, \ldots, \mathcal{T}_{i,M}]$ is an array of $M$ elements such that $M$ is the total number of columns in the table. Each row can consist of elements $\mathcal{T}_{i,\cdot}$ of heterogeneous types (*e.g.*, datetime, float, enum) with different modalities (*e.g.*, text, image), while elements in each column $\mathcal{T}_{\cdot,j}$ must be of the same format and modality.
- **Structured data**: A column $\mathcal{T}_{\cdot,j}$ is structured w.r.t. a query if it can be accessed quantitatively, such as by algebraic operations over numeric data, comparison over string labels with predefined vocabulary (*e.g.*, categorical labels), datetime functions, *etc.*.
- **Unstructured data**: A column $\mathcal{T}_{\cdot,j}$ is unstructured if a query cannot access it using standard quantitative access. Typically, such a column does not belong to a predefined taxonomy. Examples include text (*e.g.*, dialogs), images, videos, and other forms of data that usually require semantic understanding and preprocessing before performing any algebraic operations.
- **Concrete column**: A column is concrete if it already exists in the table.
- **Virtual column**: A column is virtual if it does not already exist in the table, but a query is able to operate on it. Conceptually, one needs to derive (partial rows of) these columns by processing the data from concrete columns. In our work, we bypass this step by creating such columns lazily and selectively, which is the key to achieving efficiency and performance gains.

SQL engines can perform analytic queries on databases by manipulating *structured* data in *concrete* columns. The focus of this paper is to propose a new query engine that can perform analytics on databases with both *structured* and *unstructured* data, with queries that operate over both *concrete* and *virtual* columns. Standard analytics tasks that we seek to enable over unstructured data are:

- **Conditional aggregation**: perform aggregation operations on a sub-table filtered by a condition.
- **Semantic retrieval**: collect relevant rows specified by semantic filters.
- **Abstraction and aggregation**: group the rows based on abstractions and then performs aggregation.

Since optimizing queries on structured data within concrete columns is well-studied, we focus instead on techniques for handling queries on *unstructured* data over *virtual* columns. However, the UQE implementation also supports operations over structured data within concrete columns. In the following, unless stated otherwise, the term *unstructured* databases refer to databases containing *both* structured and unstructured data.

# 3   Unstructured query language

First, we need to formally define the query language, UQL, that talks to the unstructured databases. The idea of defining a natural query language for unstructured data is not completely new (*e.g.*, Cheng et al. [11], even though UQL has richer semantics), nor is the specific syntax or design of UQL the main focus of this paper. However, we need to define the scope of queries that the engine can handle, and breakdown the semantic meaning of each clause.

## 3.1   UQL semantics

We assume a basic familiarity of SQL, upon which UQL is based. UQL can be considered to be a dialect of SQL that has augmented functionalities for handling unstructured and virtual column queries. The SQL clauses that we support in UQL, along with necessary modifications to support unstructured semantic queries, are described as follows.

**SELECT** is a mapping function that maps the operand (usually a row or collection of rows in a grouped query) to a new row of elements. In traditional SQL, this mapping is usually a subset selection over concrete columns, or algebraic operators over those columns. UQL provides additional semantic mapping capability as:

```
SELECT "the attribute specified by natural language" AS attribute_name
```

For example, one can write SELECT `"the sentiment of the movie review"` over an unstructured movie review column, and retrieve "positive" or "negative" as a structured output.

**FROM** specifies the source of the table. In SQL one can additionally specify table joins, but we limit our attention to sourcing from a single table in this paper.

**WHERE** intrinsically specifies a binary classifier over rows, which is used to retrieve a subset of the database. In addition to comparator operators on structured columns, we also allow semantic specifications in the form of:

```
WHERE "the row satisfies some natural language specifications"
```

The predicates in `WHERE` are organized in disjunctive normal form (DNF) with `AND` and `OR` syntax, so a user can arbitrarily express predicates over concrete and virtual columns.

**GROUP BY** partitions the table into groups, where rows within each group share the same attributes over the keys being grouped by. UQL allows partitioning over virtual columns via natural language:

```
GROUP BY "the abstraction criteria specified in natural language"
```

Similar to `WHERE`, one can `GROUP BY` over both concrete and virtual columns by concatenating multiple criteria, with the resulting partition corresponding to grouping by a tuple of these keys.

We also reuse other clauses from SQL including: **ORDER BY**, which simply inherits the SQL semantics to rank the resulting rows according to a specific *concrete* column. In most analytics tasks, sorting is applied over structured columns with well defined ordering comparators. **LIMIT** is applied during processing in the form of `LIMIT num_rows`, which limits the number of output rows.

**Assumptions:** We rely on the ability of an LLM to perform *intra-row* semantic understanding and analysis tasks. For example, we assume that LLMs are able to correctly judge the specification in `WHERE` for a *single* row. Similarly, LLMs should be able to extract the information specified by `SELECT` or `GROUP BY` for a *single* row. We build programmatic functionality on top of this fundamental ability of LLMs to handle analytics for large databases.

```
SELECT reason, COUNT(*) as count          SELECT agent_name, "reason to cancel"
FROM movie_reviews                        FROM airline_customer_service_log
WHERE movie_year < 2020                   WHERE "the customer asked to cancel
GROUP BY "the reason why the                     the flight"
         review is positive"              ORDER BY ticket_price
         AS reason                        LIMIT 100
```

Figure 2: Aggregation (left) v.s. Non-aggregation (right) queries written in UQL.

## 3.2 UQL queries

A UQL query is a composition of clauses that can be categorized as an aggregation or a non-aggregation, as illustrated in Figure 2. **Aggregation** queries perform a summary on groups of aggregated rows, such as COUNT the number of rows, or summarize a common attribute in a group of rows (as defined in GROUP BY above). **Non-aggregation** queries perform operations on individual rows, which usually means the rows can be processed in parallel. A UQL query will only belong to one of the above two types and we do not consider nested queries for now. The query type determines how UQE will optimize and execute the query.

## 4  Unstructured Query Engine

One straightforward way to run UQL is to use an interpreter that executes queries imperatively. Cheng et al. [11] implements an engine in this form, which is able to handle tables of relatively small size. By analogy, this is similar to executing an SQL program using a linear database scan. While it is valid, the latency and cost are prohibitive and generally prevent scaling to real world scenarios.

There are (at least) two key techniques in SQL databases for making query execution efficient:

- **Indexing**, which organizes concrete columns via data structures for fast search with sublinear cost.
- **Compilation**, which considers alternative query plans and executes the most efficient one.

UQL queries over virtual columns pose challenges in both indexing and compilation. In this section, we present effective approaches to indexing (Section 4.1) and compilation (Section 4.2) for unstructured databases, along with the implementation details of low-level primitives (Section 4.2.2).

## 4.1  Indexing

Executing traditional SQL queries over indexed columns can be made efficient by avoiding an entire database scan to find the relevant rows to process. However, for UQL queries over virtual columns, it is hard to predict or predefine an index that can enable efficient searching, since these columns are not concrete and defined via arbitrary natural language specifications.

Our first key contribution is to introduce a proxy for "indexing" that allows one to leverage the intrinsic semantic content of a virtual column to efficiently execute queries without scanning the entire database. The main idea is to use statistically sound sampling techniques to approximately retrieve relevant rows for processing. Based on the two types of queries defined in Section 3.2, we develop corresponding "indexing" counterparts.

### 4.1.1  Unbiased estimation for aggregation queries

We use a simple query to illustrate the idea of unbiased estimation to obtain a query result without scanning over an entire virtual column.

```
SELECT COUNT(*) as count FROM movie_reviews WHERE "the review is positive"
```

Given a row $\mathcal{T}_i$ (a natural language movie review), it is relatively easy for an LLM to tell whether it satisfies the WHERE condition. If we use $f : (\mathcal{T}_i, \text{cond}) \mapsto \{0, 1\}$ to represent the LLM's classification of whether row $\mathcal{T}_i$ satisfies the conditions specified in cond, then the goal is to estimate the quantity

$$\sum_{i=1}^{N} f(\mathcal{T}_i, \text{cond}) = N \sum_{i=1}^{N} \frac{1}{N} f(\mathcal{T}_i, \text{cond}) = N \mathbb{E}_{i \in \{1...N\}} [f(\mathcal{T}_i, \text{cond})] \tag{1}$$

There are many approaches that can be used to estimate the finite sum in the above equation, with different tradeoffs between bias and variance. One unbiased but potentially high variance estimator is to simply use Monte Carlo samples from a uniform distribution over $1 \ldots N$. A typical technique for reducing variance is to use importance sampling with a proposal $p$, according to

$$\mathbb{E}_{i \in \{1...N\}} [f(\mathcal{T}_i, \text{cond})] = \sum_{i=1}^{N} \frac{p_i}{N p_i} f(\mathcal{T}_i, \text{cond}) = \mathbb{E}_{i \sim p} \left[ \frac{1}{N p_i} f(\mathcal{T}_i, \text{cond}) \right] \tag{2}$$

A theoretically optimal proposal $p$ is given as follows:

---

**Algorithm 1** Stratified sampling for unbiased aggregation

---

- Embed each row $\mathcal{T}_i$ as $\vec{e}_i$, using a multi-modal embedding over the unstructured columns of $\mathcal{T}_i$.
- Cluster the embeddings $\{\vec{e}_i\}_{i=1}^N$ into $K$ disjoint groups $\{C_k\}_{k=1}^K$, $C_k \subseteq \{1 \ldots N\}$, where $k$ can be a predefined constant or automatically selected [48]. Each group has size $|C_k|$ and $\sum_{k=1}^K |C_k| = N$. We use $c : \{1 \ldots N\} \mapsto \{1 \ldots K\}$ to denote the cluster index of each row.
- Perform stratified sampling over these groups and obtain samples $S \subseteq \{1 \ldots N\}$.

Then we can obtain the following estimator for the expectation:

$$\mathbb{E}_{i \in \{1 \ldots N\}}[f(\mathcal{T}_i, \texttt{cond})] \simeq |S| \sum_{i \in S} \frac{w_i}{\sum_{j \in S} w_j} f(\mathcal{T}_i, \texttt{cond}), \text{ where } w_i = \frac{|C_{c(i)}|}{\sum_{j \in S} \mathbb{I}[c(j) = c(i)]} \quad (3)$$

---

**Algorithm 2** Online active learning for non-aggregation retrieval

---

1. Embed each row $\mathcal{T}_i$ as $\vec{e}_i$, using multi-modal embedding over the unstructured columns of $\mathcal{T}_i$.
2. Maintain $\hat{g}(i)$ that approximates $f(\mathcal{T}_i, \texttt{cond})$. Initialize $\hat{g}(i) \propto U(0, 1)$ uniformly.
3. Maintain the collection of sampled rows $S = \emptyset$ at step $t = 0$.
4. At step $t$, obtain a batch $S_t$ of samples where $S_t = \arg\max_{S_t \subseteq \{1 \ldots N\} \setminus S} \sum_{i \in S_t} \hat{g}(i) + \epsilon_{t,i}$; observe $f(\mathcal{T}_i, \texttt{cond})$ for each sample $i \in S_t$; Update $S \leftarrow S \cup S_t$.
5. Fit $\hat{g}$ with samples and corresponding observations from $S$. Go to step 4 if $|S| < B$ or return the positive samples found in $S$.

---

**Proposition 1** *The optimal proposal distribution $p$ that minimizes the variance of estimation in Eq 2 is $p_i \propto f(\mathcal{T}_i, \texttt{cond})$, which achieves zero variance.*

Prop 1 indicates that an ideal proposal should sample rows that have positive $f$ with equal probability, while sampling negative rows with zero probability. However, given that $f$ is the response of an LLM it is expensive to execute over all rows, forcing us to consider efficient approximations.

**Stratified sampling** leverages the ability to partition a population into homogeneous subpopulations. As shown in Prop 1, a good proposal $p_i$ should predict whether a row $\mathcal{T}_i$ satisfies the target property `cond`. To trade-off between cost and variance reduction, we propose Algorithm 1. In Eq 3 we normalize the importance weights $w_i$ to further reduce the estimation variance.

**Extension** The above estimator can be used for other aggregation operations such as `SUM` and `AVERAGE`, including `GROUP BY`, and allowing concrete columns as operands as well. However, some aggregations such as `MAX` does not admit such an estimator. UQE in this case can only provide estimates with greater effort. We discuss limitations in Section 7.

#### 4.1.2 Online learning for non-aggregation queries

A non-aggregation query can be viewed as a search problem where we want to find a relevant subset of rows to process. As before, we begin with a concrete example:

```
SELECT dialog_ID FROM dialogs WHERE "the customer is unhappy with the agent's manner"
```

In practice, we aim to identify as many rows as possible that satisfy the given condition while adhering to budget constraints (e.g., the total number of tokens allowed to expense). This can be formulated as an online learning problem: given the token budget (or approximately, the number of LLM calls over individual rows, denoted as $B$), we seek to balance exploration (to better understand the semantic landscape of all rows) with exploitation (to maximize recall). Drawing inspiration from Bayesian optimization, which employs a surrogate model learned on-the-fly to inform sequential decision-making [36, 19, 22], we use a cheap proxy $\hat{g}$ as a *surrogate* for $f(\mathcal{T}_i, \texttt{cond})$. At each step $t$, we re-train $\hat{g}$ with the observed data and select its maximizer to query in the next step. See Algorithm 2. Here we use random noise $\epsilon_{t,i}$ to allow some degree of exploration that decays with $t$. Compared to the typical max inner-product search method prevalent in RAG systems, we rely on the online learning to adjust the beliefs, instead of solely relying on predefined embedding similarities.

### 4.2 Compilation

Classically, the goal of a compiler is to translate a high level program to low-level machine code, maintaining *exact* execution results with improved execution speed. Such a lowering process is usually accompanied by optimizations *e.g.*, fusing, selecting the optimal instructions, and kernel optimizations. Our goal is similar: we would like to compile high-level UQL into low-level machine code, with the distinction that the "machine" is an LLM, and the "low-level code" is the orchestration of prompt calls to the LLM. Given that the primary bottleneck is the LLM API calls, we attempt to maintain the execution semantics while minimizing the cost of LLM calls, as in Figure 3.

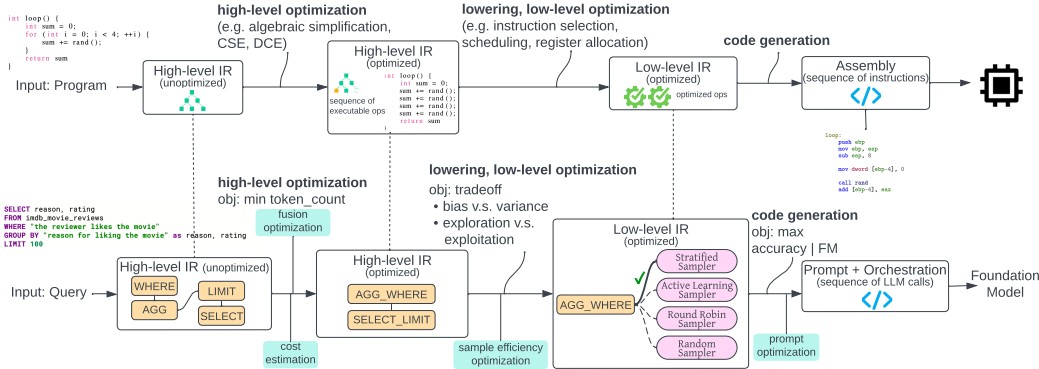

Figure 3: UQL compiler, in analogy to a typical C++ program compiler.

### 4.2.1 Planning

Lowering a query into sequences of concrete execution units is a planning problem: The action space includes the order of clause execution, as well as ways to fuse clauses to execute together. The objective is to minimize the (estimated) LLM cost. Figuring out the best decomposition and combination is usually an NP-hard problem. Fortunately, the number of clauses is very limited for a single query, so we can enumerate possible combinations of ordering and fusions with little overhead.

The outcome of planning is a specification of a sequence of kernel executions. The input and output of each kernel can be one of the following:

- **Concrete table**: a standard table with only concrete columns.
- **Stochastic table**: the outcome of unbiased sampling of a table. Importance weights will be attached to each row of the table, and the operation (*e.g.*, SUM, AVG) on this table takes weights into account.

In the following 3 sections we will explain the building blocks of the compiler, including the kernel implementation, the cost estimation and final instantiation in detail.

### 4.2.2 Kernel implementation

Each kernel is an standalone execution unit that reads and produces a (stochastic) table.

**SELECT** on structured columns is straightforward. When operating on unstructured columns, we prompt the LLM to extract semantic attributes from the input data. If several extractions share the same source column, we can also group these together into a single prompt to reduce cost.

**WHERE** takes a logical formula in disjunctive normal form, such that each conjunction can contain predicates over both unstructured and structured columns. One optimization we make in this case is to perform evaluations over the structured columns first, then simplify (*e.g.*, remove a conjunction if any of the structured column evaluates to false) the logical formula. Any remaining predicates over unstructured columns are then executed on the table filtered by predicates over structured columns.

**GROUP BY** first gathers a representative subset of rows from the table, then calls an LLM to extract a taxonomy (*i.e.*, the description of each cluster) for a cluster abstraction. Then the taxonomy is used to classify rows sampled according to the methods defined in Section 4.1. Finally, each row is classified into one of the clusters with the corresponding cluster description in the taxonomy.

**Other standard kernels** like ORDER BY are implemented as-is since they are efficient to execute.

**Kernel fusion:** Certain clauses can be fused together to achieve significant efficiency gains.

- WHERE + LIMIT can be terminated earlier for non-aggregation queries, once the number of rows specified by LIMIT are retrieved. This is particularly useful for rare event finding.
- SELECT + GROUPBY when executed together, the semantic attribute extraction of SELECT and taxonomy classification in GROUPBY can be done in the same LLM call to save cost.
- GROUPBY + WHERE can share the same sampling proposal for the aggregation queries.

When and how to fuse clauses relies on the planning technique introduced in Section 4.2.1.

### 4.2.3 Cost estimation for each kernel

We only consider the cost of calling the LLM, as this dominates the overall cost per query. Assuming the length of each row in the unstructured data is more or less uniform across rows, then the cost is proportional to the number of rows that fed to the LLM, which we use as the surrogate for estimation.

- `SELECT` maps each row, hence the cost is $|\mathcal{T}|$ for the table $\mathcal{T}$ fed to `SELECT`.
- `GROUP BY` consists of two steps, where taxonomy construction consumes a subset of the input table $\mathcal{T}$, and classification runs $|\mathcal{T}|$ LLM calls in parallel.
- `WHERE` depends on the proposal $p$. In practice we set a budget $B$ and try to minimize the variance of unbiased estimator or maximize the recall in online learning, as explained in Section 4.1.
- Whenever clauses are fused together, each implementation is responsible for providing a reasonable cost estimate. For example when `SELECT` and `GROUP BY` are fused, the estimated cost is the same as `GROUP BY` alone, as the classification stage of `GROUP BY` shares the input tokens with `SELECT`.

### 4.2.4 Instantiation of kernels

The last step of compilation is to generate the machine specific code (*e.g.*, x86 assembly code) from the intermediate representations (IR). For UQE, this is the process of generating the LLM-specific prompts. For example, when GPT is deployed as the "machine", a system prompt like "You are a helpful assistant" will be added to the queries. This step also sets the correct context (*e.g.*, the correct structured/unstructured column to associate to, the description of the databases) for the LLM. When such information is not available, one can also leverage the LLM to provide a good suggestion.

## 5 Related work

While the unstructured data analytics engine is relatively new, there are several related works in the context of unstructured data query and analysis. Approaches like pattern or regexp matching [20] is scalable but not feasible for complex semantic reasoning. RAG [24, 18] based approaches rely on the retrieval quality and is not directly suitable for aggregation queries over entire database. LLMs [4, 2, 33] depict the ability of table analytics [15] to some extent [9, 25], but are still not reliable for large unstructured database analytics yet.

Our work is closely related to neural symboilic approaches for unstructured data analytics. Early attempts in this line aim to design specialized neural architectures with inductive biases (e.g., attention) to capture a particular form of operation (e.g., filtering a list of objects based on a natural language predicate by their attention scores) [43, 29, 3]. Those differentiable neural "operators" can then be chained together to model more compositional queries, and trained end-to-end using gradient descent. Another direction, in line with our work, is to augment symbolic programs with learnable operators parameterized by neural networks [10]. Those programs are often modeled as discrete latent variables, which can be hard to optimize. In contrast, UQE leverages predictions from LLMs as supervision to train an efficient proxy query model in an online fashion. Similar to UQE, some recent work [11, 38] also adopts LLMs as fuzzy query operators. However, the generated programs treat LLMs as an UDF in a SQL program, which can be very expensive to execute on large databases. Our UQE implements similar but augmented semantics with the focus on the cost efficiency and scalability. Liu et al. [26] optimizes a similar query engine from the system perspective like cache optimization and deduplication, while our work mainly considers algorithmic improvements and is considered as an approximate query engine [28]. These system and algorithm optimizations are actually orthogonal and can be beneficial to jointly consider both for future works.

In a distantly related topic, text2SQL [47, 46, 21, 35] also leverages models talking to databases, but is mainly for semantic parsing purpose. While it also leverages the advances in LLMs [44, 16, 31, 37], the execution is still on pure SQL and thus is not suitable for unstructured databases. There are also works on leveraging formal query languages to better query LLMs [34, 6], with the focus on controllability of the LLM itself rather than performing analytics on external unstructured data.

## 6 Experiments

We benchmark the accuracy and incurred cost of UQE on multimodal unstructured data analytics tasks, with the goal to show and understand when and why UQE can improve accuracy while keeping the cost low. Since the unstructured database analytics is a relatively new task, we construct and compare against several baseline approaches, on a set of tasks created from existing datasets.

Table 1: Conditional aggregation results on benchmark datasets. We report the relative error and the average cost per query. *gpt-4o is 50% cheaper than gpt-4-turbo so we double its budget of tokens; [†] we use claude-3-haiku as the backend LLM.

| Benchmarks | Conditions | Methods | | | |
|---|---|---|---|---|---|
| | | lc-gpt-4-turbo | lc-claude-3-opus | UDF[†] | UQE [†] |
| IMDB | sentiment_positive | $49.02\% \pm 21.23\%$ | $56.05\% \pm 14.69\%$ | $13.67\% \pm 6.24\%$ | $\mathbf{5.75\% \pm 3.43\%}$ |
| | Average cost per query | $0.37 | $0.61 | $0.01 | $0.01 |
| ABCD | account_access | $69.25\% \pm 32.82\%$ | $27.28\% \pm 17.05\%$ | $18.99\% \pm 9.85\%$ | $\mathbf{11.75\% \pm 9.78\%}$ |
| | single_item_query | $78.42\% \pm 9.36\%$ | $23.39\% \pm 16.14\%$ | $26.95\% \pm 22.16\%$ | $\mathbf{12.32\% \pm 10.53\%}$ |
| | Average cost per query | $0.38 | $0.63 | $0.01 | $0.01 |
| AirDialog | book | $47.58\% \pm 15.24\%$ | $54.40\% \pm 13.37\%$ | $10.15\% \pm 7.64\%$ | $\mathbf{4.98\% \pm 2.26\%}$ |
| | no_flight | $47.92\% \pm 21.62\%$ | $53.88\% \pm 15.77\%$ | $21.08\% \pm 16.78\%$ | $\mathbf{8.78\% \pm 8.12\%}$ |
| | no_reservation | $50.54\% \pm 21.86\%$ | $57.49\% \pm 13.08\%$ | $21.19\% \pm 12.10\%$ | $\mathbf{7.23\% \pm 5.40\%}$ |
| | Average cost per query | $0.21 | $0.36 | $0.01 | $0.01 |
| | | lc-gpt-4o | lc-gpt-4-turbo | UDF-gpt-4o | UQE-gpt-4o |
| Clevr | obj_count < 4 | $22.46\% \pm 19.35\%$ | $48.15\% \pm 41.52\%$ | $31.04\% \pm 25.15\%$ | $\mathbf{9.55 \pm 8.55\%}$ |
| | # spheres > 3 | $35.72\% \pm 14.95\%$ | $91.09\% \pm 20.08\%$ | $19.35\% \pm 13.81\%$ | $\mathbf{15.14\% \pm 10.71\%}$ |
| | Average cost per query | $0.33* | $0.33 | $0.20 | $0.20 |

Table 2: Semantic retrieval results on several benchmark dataset. We report the F1 score of the retrieved rows and the average cost per query. We run 8 independent queries and report the average F1 and its standard deviation. The result of MIPS is deterministic, so no standard deviation is reported.

| Benchmarks | Conditions | Methods | | | | |
|---|---|---|---|---|---|---|
| | | lc-gpt-4-turbo | lc-claude-3-opus | UDF | MIPS | UQE |
| IMDB | sentiment_positive | $0.397 \pm 0.041$ | $0.556 \pm 0.066$ | $0.505 \pm 0.030$ | $0.875$ | $\mathbf{0.978 \pm 0.003}$ |
| | Average cost per query | $0.38 | $0.63 | $0.02 | $\simeq$ $0 | $0.02 |
| ABCD | account_access | $0.045 \pm 0.033$ | $0.080 \pm 0.023$ | $0.076 \pm 0.017$ | $\mathbf{0.961}$ | $0.940 \pm 0.019$ |
| | single_item_query | $0.023 \pm 0.021$ | $0.082 \pm 0.030$ | $0.065 \pm 0.017$ | $0.266$ | $\mathbf{0.935 \pm 0.006}$ |
| | Average cost per query | $ 0.76 | $1.23 | $0.03 | $\simeq$ $0 | $0.03 |
| AirDialog | book | $0.327 \pm 0.0667$ | $0.585 \pm 0.025$ | $0.342 \pm 0.031$ | $0.930$ | $\mathbf{0.979 \pm 0.010}$ |
| | no_flight | $0.066 \pm 0.037$ | $0.228 \pm 0.068$ | $0.144 \pm 0.034$ | $0.867$ | $\mathbf{0.928 \pm 0.018}$ |
| | no_reservation | $0.156 \pm 0.075$ | $0.297 \pm 0.043$ | $0.145 \pm 0.042$ | $0.965$ | $\mathbf{0.969 \pm 0.004}$ |
| | cancel | $0.006 \pm 0.009$ | $0.013 \pm 0.008$ | $0.013 \pm 0.009$ | $0.066$ | $\mathbf{0.741 \pm 0.205}$ |
| | Average cost per query | $0.43 | $ 0.74 | $0.01 | $\simeq$ $0 | $0.01 |
| Clevr | obj_count < 4 | $0.058 \pm 0.026$ | $0.066 \pm 0.023$ | $0.093 \pm 0.031$ | $0.023$ | $\mathbf{0.850 \pm 0.025}$ |
| | # spheres > 3 | $0.037 \pm 0.027$ | $0.099 \pm 0.023$ | $0.089 \pm 0.017$ | $0.145$ | $\mathbf{0.633 \pm 0.177}$ |
| | Average cost per query | $0.38 | $0.21 | $0.08 | $\simeq$ $0 | $0.08 |

**Baselines:** We design the following baselines for comparison

- **lc-LLM** denotes the long-context LLMs that can directly take a subset of database and a natural language question as input, and produce the desired analysis. We mainly evaluate against several model families, including GPT-4 [2] and Claude-3 [4]. Of course, when evaluating the lc-LLM based approaches, we use the natural language instead of UQL as the prompt.
- **RAG-based** can be applied to some non-aggregation queries, such as semantic retrieval. For the retrieval part we use max inner-product search (MIPS) on top of the same embeddings that are used by UQE, for a controlled experiment.
- **UDF** simply treats the LLM calls as User-defined function of an SQL engine, with the same budget as UQE by default. This approach will not have the advanced sampling / search algorithm as used in UQE, which also serves as an ablation for the effectiveness of our UQE.

**Datasets:** We evaluate different approaches on common analytical tasks in three widely used application domains. We use the datasets that were previously created for discriminative tasks, as these datasets contain both the unstructured columns and the structured ones (the labels in the corresponding dataset). We then hide these structured label columns and perform analytical tasks on the unstructured columns, where these hidden structured columns will be used to compute the ground-truth. The text based tasks include IMDB [27] movie reviews, customer service dialogs including Action-Based Conversations Dataset (ABCD [8]) and AirDialog [41], and image based Clevr [23] dataset. Please refer to Appendix C.1 for more information.

**Setup:** We use voyage-2 [1] to embed the text-based unstructured columns, and Vertex [40] for multimodal embeddings. For budget constraint queries, we allow different approachces to access at most 128 rows in the database by default.

Table 3: Conditional abstraction and aggregation.

| Benchmarks | Metrics | Methods | | |
|---|---|---|---|---|
| | | lc-gpt-4-turbo | lc-claude-3-opus | UQE-claude-3-haiku |
| AirDialog | EMD↓ | $0.143 \pm 0.034$ | $0.121 \pm 0.014$ | $\mathbf{0.111 \pm 0.019}$ |
| | cost | \$0.21 | \$0.37 | **\$0.04** |
| ABCD | account_access-EMD↓ | $0.154 \pm 0.031$ | $0.113 \pm 0.010$ | $\mathbf{0.110 \pm 0.016}$ |
| | single_item_query-EMD↓ | $0.031 \pm 0.034$ | $0.011 \pm 0.006$ | $\mathbf{0.005 \pm 0.002}$ |
| | cost | \$0.34 | \$ 0.56 | \$ 0.07 |

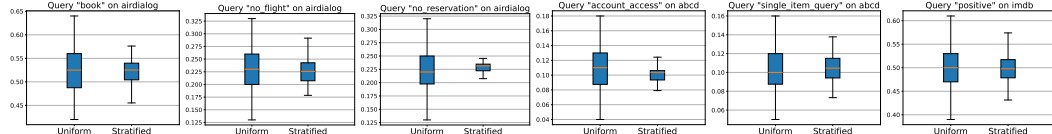

Figure 4: Variance of different sampling approaches for aggregation queries over 3 text datasets.

## 6.1 Main results

We run queries on different datasets by instantiating the template shown in each of the sections below. The exact natural language and UQL queries can be found in Appendix D and more information including statistics of conditions we used for query and hyperparameters (for UQE we simply use the default hyperparameters for sampling and online learning) can be found in Appendix C.

### 6.1.1 Conditional aggregation

This task provides aggregated statistics over databases with specified conditions, with the template as:

```
SELECT COUNT(*) FROM {table} WHERE {"satisfies natural language specified condition"}
```

We report the relative estimation error (*i.e.*, $|\text{predict} - \text{true\_count}|/\text{true\_count}$) and its standard deviation in Table 1. For lc-LLM baselines we estimate the count based on groups of unbiased data samples that fed into the prompt.

For text based aggregation we use claude-3-haiku as the backbone model, where UQE deploys $10\times$ reduction in relative errors while reducing the cost by a factor of $20\times$ or more. For the image dataset, since only limited set of LLMs are capable right now, we use gpt-4o as the backbone, and compare with lc-LLM baselines. Thanks to the improved sampling method in UQE, the same gpt-4o consistently achieves improved performance out-of-the-box. To verify this, we feed one image at a time to gpt-4o and manually aggregate the count, the estimation error would be $17.10\% \pm 13.95$ and $19.35\% \pm 13.81\%$ for the two queries of Clevr, which is twice higher than UQE in the worst case.

### 6.1.2 Semantic retrieval

This task filters rows in databases that satisfy specified conditions, with the template as:

```
SELECT * FROM {table} WHERE {"satisfies natural language specified condition"} LIMIT B
```

While we limit the output size to be B = 256 to keep the total cost within a reasonable budget. The challenging scenarios are when the number of rows that satisfy the predicate is few (*i.e.*, "rare event finding"). Table 2 shows similar sets of comparison, but the metric is F1 score which evaluates the quality of SELECT-ed rows. Overall UQE (with claude-3-haiku as backbone LLM) consistently achieves comparable or better performance than the baseline methods. MIPS which uses the same embedding of unstructured data as UQE, has high variance across different types of queries. The queries such as "dialogs with account access issues" would be very suitable for MIPS as the embedding similarity is able to capture that well. For queries involving reasoning (*e.g.*, find the images with less than 4 objects), it is pretty hard for pretrained embeddings to express this.

### 6.1.3 Abstraction and aggregation

This task abstracts the intrinsics of each row, and then performs semantics-based GROUP BY, grouping the common intrinsics across all rows. Finally, it provides aggregated statistics over each group:

```
SELECT derived_attribute, COUNT(*) FROM {table}
GROUP BY {"extract an abstract intrinsic attribute specified in natural language"}
AS derived_attribute LIMIT 10
```

The challenging problems in this task are (i) building a taxonomy with good coverage, and (2) bias and variance reduction for groups with small population. The result of this query is a list of

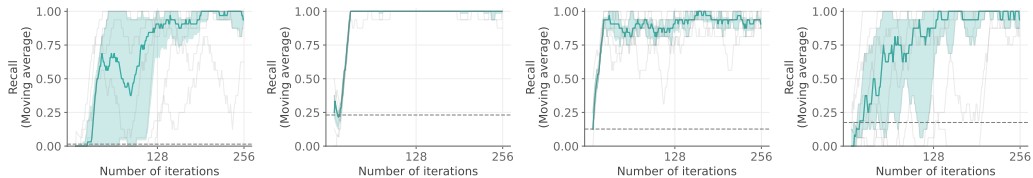

Figure 5: Recall (moving average with window size 16) against the number of iterations on (from left to right) AirDialog with condition {cancel, no_flight} and Clevr with {obj_count < 4, #spheres > 3}. Colored lines and shades denote median and interquartile ranges across 8 independent queries and gray lines denote individual queries. The gray dashed lines denote the fraction of the positive population in the entire dataset.

Table 4: Quality at different compute budget $B$.

| | Budget $B$ | 256 | 128 | 64 | 32 |
|---|---|---|---|---|---|
| Retrieval | latency(s) | 38.08 | 21.61 | 11.11 | 5.84 |
| | F1 score | 0.978 | 0.974 | 0.921 | 0.828 |
| Aggregation | latency(s) | - | 5.83 | 4.28 | 2.93 |
| | Error | - | 5.75% | 6.84% | 8.29% |

Table 5: Error of the aggregation operation on IMDB dataset under different budget $B$ .

| Budget $B$ | 512 | 256 | 128 | 64 | 32 |
|---|---|---|---|---|---|
| UDF | 8.11% | 8.35% | 13.67% | - | - |
| UQE | - | - | 5.75% | 6.84% | 8.29% |

tuples of derived attributes and their number of occurrences in the dataset. We use the earth mover's distance (EMD [30]) as the evaluation metric to compare the extracted tuples and ground-truth tuples. The distance between a pair of attributes is defined by one minus the cosine similarity of their text embeddings. We can see from Table 3 that UQE consistently outperforms baselines while achieving much lower cost. We also show in Appendix C with more qualitative results comparisons.

## 6.2 Ablation studies

We study the effectiveness of UQE for aggregation queries in Section 6.2.1 and non-aggregation queries in Section 6.2.2, and the quality/cost trade-off of UQE in Section 6.2.3. In appendix we provide more results on other modalities C.3, consistency C.4 and latency C.5.

### 6.2.1 Variance of different sampling approaches for aggregation queries

To decouple the variance introduced by the algorithm and the bias introduced by the LLM based predictors, here we use the ground-truth label as the predictive result and focus on the effectiveness of variance reduction. Figure 4 shows the box plot of different sampling methods. We can see using stratified sampling over the embeddings of unstructured content achieves significant lower variance compared to the uniform random sampling. Also both of these achieve similar expected values, which also justifies the correctness or unbiasedness.

### 6.2.2 Efficiency of online learning for non-aggregation queries

We show the effectiveness of the online learning in terms of the recall as a function of the iteration steps in Figure 5. Compared to the dashed line in the figure which indicates the results of uniform random sampling, the online learning can achieve significant boost in terms of the recall. While for some queries the variance at early iterations can be high, these all converge well in the end.

### 6.2.3 Trade-off between cost/latency and accuracy

Generally the larger compute budget $B$ the better quality UQE will get, and we verify this in Table 4 5. UQE can achieve pretty good quality even with very low budget, and notably compared to the baseline, it achieves similar quality with 16x reduction of the compute needed. We show more results in Section C.5 regarding the compute efficiency.

## 7 Conclusion

This paper proposed an unstructured query engine that leverages 1) the flexibility of LLMs for data understanding; 2) the advances in sampling and online learning for efficient data scanning; 3) and the compiler that bridges these algorithmic workflows with LLMs. We demonstrated its efficiency and accuracy over three analytic tasks on four datasets with two different modalities. However the current work is still very limited in terms of 1) the semantics it lacks, including table join and other types of aggregations; 2) an automated selection of LLMs and sampling configurations; 3) and scaling to even larger databases. We hope to investigate these further in future works.

## Acknowledgments and Disclosure of Funding

We thank Carsten Binnig, Howie Xu, Ras Bodik, Xinyi Chen, and the anonymous reviewers for providing helpful discussion and suggestions.

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

# A Proof

**Proposition**: *The optimal proposal distribution $p$ that minimizes the variance of estimation in Eq 2 is $p_i \propto f(\mathcal{T}_i, \texttt{cond})$. The variance gets 0 with this proposal.*

Let's simplify the notation a bit and use $f(x)$ for $f(\mathcal{T}_i, \texttt{cond})$ and prove the variance reduction in general cases for binary function $f$. For the simplicity let's omit the constant $|\mathcal{T}|$ and focus on the estimation of the expectation term. If we come up with a new proposal distribution $p : \mathcal{T} \mapsto [0, 1]$ where $\sum_{x \in \mathcal{T}} p(x) = 1$, then we get a new estimator in the following form:

$$\mathbb{E}_{x \sim q}[f(x)] = \sum_{x \in \mathcal{T}} [p(x) \frac{q(x)}{p(x)} f(x)] = \mathbb{E}_{x \sim p}[\frac{q(x)}{p(x)} f(x)] \tag{4}$$

We hope this new estimator would have lower variance. Let's define $u(x) = \frac{q(x)}{p(x)} f(x)$ for the ease of notation, and look at its variance first:

$$Var_p(u(x)) = \mathbb{E}_p[u^2(x)] - \mathbb{E}_p^2[u(x)] \tag{5}$$

Since our estimator is unbiased, $\mathbb{E}_p^2[u(x)] = E_q^2[f(x)]$ and thus has nothing to do with $p$, let's focus on minimizing the first term. More specifically we have the optimization as:

$$\begin{aligned} \min_p \quad & \mathbb{E}_p[u^2(x)] \\ s.t. \quad & p(x) \geqslant 0, \forall x, \\ & \sum_{x \in \mathcal{T}} p(x) = 1 \end{aligned} \tag{6}$$

Let:

$$L(p, \{\lambda_x\}, \lambda_2) = \sum_{x \in \mathcal{T}} \frac{(q(x)f(x))^2}{p(x)} - \sum_{x \in \mathcal{T}} \lambda_x p(x) + \lambda_2(\sum_{x \in \mathcal{T}} p(x) - 1) \tag{7}$$

and we can find the saddle point of $\min_p \max_{\lambda_x \lambda_2}(L(p, \{\lambda_x\}, \lambda_2))$ using K.K.T condition.

$$\begin{cases} -\frac{(q(x)f(x))^2}{p(x)^2} - \lambda_x + \lambda_2 = 0, \forall x \in \mathcal{T} \\ \lambda_x p(x) = 0, p(x) \geqslant (0), \forall x \\ \lambda_2 \sum_x (p(x) - 1) = 0, \sum_x p(x) = 1 \end{cases}$$

and we can get the optimal solution of

$$p(x) = \frac{q(x)f(x)}{E_q[f(x)]} \tag{8}$$

and put it back into Eq(1) we can see the optimal variance would be

$$\begin{aligned} Var_p(u(x)) &= \sum_x \frac{(q(x)f(x))^2}{\frac{q(x)f(x)}{\mathbb{E}_q[f(x)]}} - \mathbb{E}_p^2[u(x)] \\ &= \mathbb{E}_q[f(x)] \sum_x q(x)f(x) - \mathbb{E}_q^2[f(x)] = 0 \end{aligned} \tag{9}$$

which means one sample would be good enough! In our context, $q(x)$ is usually just a constant (e.g., $q(x) = \frac{1}{\mathcal{T}}$ for the [Case Count]), and $f(x) \in \{0, 1\}$. In this case, a simplified optimal proposal would be:

$$p(x) \propto f(x) \text{ and the partition function is } \mathbb{E}_q[f(x)] \tag{10}$$

or in another word, ideally we should have zero-chance to sample from regions where $f(x) = 0$, and have equal chances to sample where $f(x) = 1$.

# B   UQL specifications

## B.1   Tokenizer

We use the following pattern matching to tokenize the UQL programs/queries.

```python
import ply.lex as lex
reserved = {
    'select' : 'SELECT',
    'from' : 'FROM',
    'where': 'WHERE',
    'as' : 'AS',
    'limit': 'LIMIT',
    'group': 'GROUP',
    'order': 'ORDER',
    'by': 'BY',
    'to': 'TO',
    'and': 'AND',
    'or': 'OR',
    'count': 'COUNT',
    'avg': 'AVG',
    'sum': 'SUM',
    'desc': 'DESC',
}
tokens = [
  'SEPARATOR',
  'ALL',
  'NL_LITERAL',
  'VAR_NAME',
  'TABLE_URL',
  'COMPARE_OPERATOR',
  'INTEGER',
  'FLOAT',
  'LEFT_PARENTHESIS',
  'RIGHT_PARENTHESIS',
] + list(reserved.values())
t_SEPARATOR = r','
t_ALL = r'\*'
t_NL_LITERAL = r'"((?:\\.|[^"\\])*)"'
t_COMPARE_OPERATOR = r'(<>|>=|<=|!=|>|<|=)'
t_INTEGER = r'[-]?\d+'
t_FLOAT = r'[+-]?[0-9]*\.[0-9]+'
t_LEFT_PARENTHESIS = r'\('
t_RIGHT_PARENTHESIS = r'\)'
def t_VAR_NAME(t):
  r'[a-zA-Z_][a-zA-Z_0-9]*(\.[a-zA-Z_][a-zA-Z_0-9]*)*'
  t.type = reserved.get(t.value.lower(), 'VAR_NAME')
  return t
```

## B.2   Grammar

Below we show the context free grammar of the UQL. Note that this represents a subset of the "natural query" analogy to SQL, and we leave other clauses like table join into the future work.

```
============================
|                          |
|       UQL grammar        |
|                          |
============================

uql_query : select_clause from_clause
          | select_clause from_clause optional_clause_combo
```

```
optional_clause_combo : optional_clause_combo optional_clause
                      | optional_clause

optional_clause : limit_clause
                | to_clause
                | where_clause
                | group_by_clause
                | order_by_clause

select_clause : SELECT select_expression

select_expression : select_expression SEPARATOR select_literal
                  | select_literal

select_literal : ALL
               | variable_literal
               | nl_literal
               | aggregation
               | INTEGER

aggregation : agg_op LEFT_PARENTHESIS VAR_NAME RIGHT_PARENTHESIS
            | agg_op LEFT_PARENTHESIS ALL RIGHT_PARENTHESIS
            | agg_op LEFT_PARENTHESIS VAR_NAME RIGHT_PARENTHESIS AS VAR_NAME
            | agg_op LEFT_PARENTHESIS ALL RIGHT_PARENTHESIS AS VAR_NAME

agg_op : AVG
       | COUNT
       | SUM

variable_literal : VAR_NAME
                 | VAR_NAME AS VAR_NAME

nl_literal : NL_LITERAL
           | NL_LITERAL AS VAR_NAME

from_clause : FROM VAR_NAME

where_clause : WHERE where_expression

where_expression : where_expression AND predicate
                 | where_expression OR predicate
                 | predicate

group_by_clause : GROUP BY group_by_expression

group_by_expression : group_by_expression SEPARATOR group_by_literal
                    | group_by_literal

group_by_literal : variable_literal
                 | nl_literal

order_by_clause : ORDER BY order_by_expression
                | ORDER BY order_by_expression DESC

order_by_expression : order_by_expression SEPARATOR order_by_literal
                    | order_by_literal

order_by_literal : VAR_NAME
                 | NL_LITERAL
                 | INTEGER

predicate : NL_LITERAL
          | VAR_NAME COMPARE_OPERATOR NL_LITERAL
          | VAR_NAME COMPARE_OPERATOR INTEGER
```

| Dataset | Conditions | percentage of occurrence in the data |
|---|---|---|
| Airdialog | book | 51.40% |
| | cancel | 1.46% |
| | no_flight | 23.08% |
| | no_reservation | 23.89% |
| ABCD | single_item_query | 10.41% |
| | account_access | 10.44% |
| IMDB | positive_review | 50% |
| Clevr | `obj_count < 4` | 12.65% |
| | `# spheres > 3` | 17.53% |

Table 6: Dataset statistics

```
                | VAR_NAME COMPARE_OPERATOR FLOAT

limit_clause  :  LIMIT  INTEGER

to_clause  :  TO  VAR_NAME
```

## C   Experiments

### C.1   Datasets

We evaluate different approaches on common analytical tasks in three widely used application domains. We use the datasets that were previously created for discriminative tasks, as these datasets contain both the unstructured columns and the structured ones (the labels in the corresponding dataset). We then hiden these structured label columns and perform analytical tasks on the unstructured ones, where these hidden structured columns will be used to compute the groundtruth.

- **User review mining**. We use IMDB [27] for semantic analysis of user sentiment. The entire dataset contains 50K highly polar movie reviews with positive and negative sentiment labels.
- **Goal-oriented (customer service) dialogue systems**. We use two datasets for this category, including 1) Action-Based Conversations Dataset (ABCD [8]) for intent identification, which has 10,042 dialogs with 10 distinct user intents requiring unique sequences of actions constrained by policies to achieve task success; and 2) AirDialog [41] for conversation outcome understanding. It contains 402,037 goal-oriented conversation on flight booking with 5 possible ground-truth states.
- **Image understanding**. We use Clevr [23] dataset for multimodal data understanding and retrieving. Specifically we use the val split of the dataset, which contains 15,000 images of objects containing different number of cylinders, cubes and spheres with different sizes/colors. We down-sample the image to size no more than $128 \times 128$, and only feed the images to the LLMs while holding out scene metadata for evaluation only.

Also see Table 6 for the detailed statistics of different query conditions. Rare events like `cancel` in AirDialog is typically challenging to find and aggregate on.

### C.2   Parameter and experiment setup

For the embeddings, we use the voyage-2 for text and for images, we use Google Vertex API with dimensionality of 512. We preprocess the embeddings for all the datasets and keep them static during the queries.

For the aggregation queries, we use faiss [2] to cluster the embeddings into 10 groups, and perform stratified sampling on top.

For the online learning setting for non-aggregation queries, in our experiment we simply set $\epsilon_{t,i}$ to be 0. We start training the function $g$ when we collect at least one possible and one negative example labeled by the LLM. Then after every minibatch of samples collected, we train $g$ via linear logistic regression and simply leverage sklearn for that.

Other parameters that might matter include: the sampling budget $B$ for aggregation queries is 128 and for non-aggregation queries it is 256. For group-by queries the UQE needs a step in building the taxonomy, where the budget we use for that is 16.

---

[2]https://github.com/facebookresearch/faiss

Table 7: Retrieval F1 score on AudioMnist dataset.

| Query | UDF-gemini | MIPS | UQE-gemini |
|---|---|---|---|
| "Three" | $0.109 \pm 0.022$ | 0.445 | $\mathbf{0.839 \pm 0.135}$ |
| "A number whose square equals to itself" | $0.259 \pm 0.068$ | 0.039 | $\mathbf{0.922 \pm 0.024}$ |
| "The minimum prime number" | $0.107 \pm 0.044$ | 0 | $\mathbf{0.917 \pm 0.025}$ |

Table 8: Performance of UQE with different LLM backends.

| Task | UDF | UQE-haiku (in the main paper) | UQE-gpt4o-mini-0718 |
|---|---|---|---|
| Retrieval | $0.505 \pm 0.030$ | $0.978 \pm 0.003$ | $0.956 \pm 0.010$ |
| Aggregation | $13.67\% \pm 6.24\%$ | $5.75\% \pm 3.43\%$ | $6.33\% \pm 4.71\%$ |

We set these parameters based on educated guess and keep them as default across all the queries over all the datasets.

### C.3 Audio modality

We use the audio MNIST [5] data for this experiment. This dataset contains 30k wav files from 60 speakers pronouncing digits 0-9. We perform the audio semantic retrieval experiments. The query is first converted to audio space using TTS and the corresponding audio embedding is used for MIPS search. We use Gemini Pro 1.5 as the backend model, as it supports the audio inputs nicely. As is shown in Table 7, the proposed UQE consistently does better than alternatives, and it also allows complex queries that require reasoning, while embedding based MIPS is limited to certain types of queries.

### C.4 Consistency of the execution results

While UQE is able to leverage the foundation models to analyze the unstructured databases directly, the execution result is not deterministic due to the nature of the stochasticity of the algorithm and the LLMs themselves. In our paper we aim at reducing the variance so as to improve the consistency. However when the backend LLM gets updated, it might also cause the potential inconsistency. Below we analyze the effect of different LLM backends.

Table 8 shows the effect on the IMDB dataset, where we see little variation. Actually the difference caused by the model switching is much lower than using a worse query engine. Of course, this behavior shift would be task related, but with the advances of LLMs we believe this variation would converge and be more stable to different prompts in the future.

### C.5 Latency

We report the runtime of UQE with claude-3-haiku as backbone, and lc-gpt-4-turbo as the baseline method in Table 9. We can see UQE achieves low latency in aggregation operations, but higher latency in retrieval. This is due to the online update and re-evaluation of the $g$ function described in Section 4.1.2. The experiments were run on MacBook Pro CPU, so we expect this bottleneck would be alleviated with better engineered system, which we will focus in our future works.

### C.6 Group By qualitative results

For `single_item_query` in ABCD, the items mentioned in the dialogs found by UQE is:

```
[['boots' '338']
 ['jacket' '282']
 ['jeans' '268']
 ['shirt' '190']]
```

For `account_access` in ABCD, the issues mentioned in the dialogs found by UQE is:

```
[['Forgot Password' '457']
 ['Forgot Username' '406']
 ['Lost phone for two-factor authentication' '362']]
```

Which is very close to the ground truth (recover_username, reset_2fa, recover_password).

For airdialog, the outcomes found by UQE is

Table 9: Runtime (in seconds) comparison for different types of queries over different benchmarks.

| Methods | Conditional aggregation | | | | Semantic Retrieval | | | |
|---|---|---|---|---|---|---|---|---|
| | Clevr | ABCD | IMDB | Airdialog | Clevr | ABCD | IMDB | Airdialog |
| UQE-claude-3-haiku | 3.13 | 3.34 | 5.83 | 3.85 | 46.00 | 41.20 | 38.08 | 67.14 |
| lc-gpt-4-turbo | 28.06 | 4.72 | 4.37 | 3.23 | 63.38 | 10.10 | 20.61 | 23.06 |

```
[['Flight Ticket Booked' '199383']
 ['No Flights Available' '100884']
 ['No Reservation Found' '80775']
 ['Flight Reservation Cancelled' '51937']]
```

Where the ground truth has one more additional outcome (cancel). But since the percentage of cancellation is very small, it is expected that this might be missing from the group by abstraction when number of occurs are very limited.

## D  Prompts

### D.1  Prompts for lc-LLMs

#### D.1.1  Task: Conditional aggregation

**IMDB dataset**

**System prompt**

```
Read the following movie reviews, and categorize them into either positive
or negative class, depending on the sentiment of the review.If the movie
has a mixed sentiment, try your best to classify into positive or negative
class based on the overall sentiment.In the end, please just output a
single number, which is [the total number of positive reviews]
```

**User prompt**

```
Below are the reviews:
[Review 0]: Some TV programs continue into embarrassment (my beloved 'X-
    ↪ Files' comes to mind.)...
[Review 1]: The tale of the titular Adam (Mark O' Halloran) and Paul (Tom
    ↪ Murphy), ...
```

**ABCD dataset**

**System prompt**

```
The following dialogs between a customer service agent and a customer.
Dialogs start by headers such as **Dialog 1**, **Dialog 2**, and so on.
Your task is to classify whether the dialog content is about the theme
"account access issue".  Then count how many dialogs are talking about this
theme.  In the end, output the count as a single number.
Here is more detailed explanation about Theme "<THEME>".  Be sure to use
this information when you classify.
Theme "<THEME>" dialogs content is about THEME_EXPLANATION.  Please perform
thorough analysis for each of the dialog.  In the end, please **only**
output a single number, which is [the total number of dialogs] that talks
about the theme "account access issue".

<THEME> = account access issue
<THEME> = requesting detailed specifications of a certain item sold on the
website

<THEME_EXPLANATION> = the customer could not access the account and was
locked out, such as couldnt́ recall their username, couldn't perform
two-factor authentication, or forgot their password and couldn't access
their account
```

<THEME_EXPLANATION> = the customer needed help with the detailed
specifications of a certain item sold on the website, such as inquiries
about detailed information of a specific retail item about materials,
whether it shrinks, stock availability, etc., but NOT inquiries about
promotions, order status, shipping status, questions about how to use the
website to purchase an item, or difficulties on using the website to add to
cart or purchase an item, or inquiries about subscription

**User prompt**

Below are the dialogs:
**Dialog 0**:
[agent]:  Hello, how can i help you today
[customer]:  Hello my name is Alessandro Phoenix and I need to make sure
the shipping cost is included on my order
...

**AirDialog dataset**

**System prompt**

The following are dialogs between a airline ticketing agent and a customer.
Dialogs start by headers such as **Dialog 1**, **Dialog 2**, and so on.
The outcome of the dialog will be one of the following 5 categories:
[book]:  the agent has booked a flight for the customer (not including the
flight change);
[cancel]:  the agent canceled the existing valid reservation for the
customer;
[no_reservation]:  the customer wants to change or cancel the flight 1037
but there is no valid reservation under this customer;
[no_flight]:  the customer aims to book a flight from departure to
destination but finds no flights between departure and destination;
Your task is to count how many dialogs have outcome <OUTCOME>.  Please
**only** output a single number, which is [the total number of dialogs]
that satisfied the above requirements.

<OUTCOME> = [book]
<OUTCOME> = [cancel]
<OUTCOME> = [no_reservation]
<OUTCOME> = [no_flight]

**User prompt**

Below are the dialogs:
**Dialog 0**:
customer: Hello.
agent: Hello, how can I help you today?
customer: Can you please find a flight from DFW to SEA?
...
**Dialog 1**:
customer: Hi, I am Melissa Thompson.
agent: Hello, how may I support you today?
customer: I want to celebrate Thanks giving day 11/23 with my friends
at New York. Can you book a ticket for me?
...

**Clevr dataset**

**Task: Conditional aggregation**

**System prompt**

Please read the following images, and count how many of them show that
<CONDITION>.

```
<CONDITION> = there are less than 4 objects in the image
<CONDITION> = there are more than 3 spheres in the image
```

**User prompt**

```
image\_0: <base64\_encoded\_image>
image\_1: <base64\_encoded\_image>
...
Please output a single number, which is the total number of images that
    ↪ satisfy the condition.
```

### D.1.2   Task: Semantic retrieval

**IMDB dataset**

**System prompt**

```
Read the following movie reviews, and list the indices of reviews with
positive sentiment.  If the movie has a mixed sentiment, try your best to
classify into positive or negative class based on the overall sentiment.In
the end, please only output a list of indices in the format of [review_3,
review_7, ...]
```

**User prompt**

```
Below are the reviews:
[Review 0]: Some TV programs continue into embarrassment (my beloved 'X-
    ↪ Files' comes to mind.)...
[Review 1]: The tale of the titular Adam (Mark O' Halloran) and Paul (Tom
    ↪ Murphy), ...
```

**ABCD dataset**

**System prompt**

```
The following are dialogs between a customer service agent and a customer.
Dialogs start by headers such as **dialog_1**, **dialog_2**, and so on.
Your task is to find out the dialogs where <CONDITION>.  Please only output
a list of indices in the format of [dialog_3, dialog_7, ...]
```

```
<CONDITION> = the customer could not access the account and was locked
out, such as couldnt́ recall their username, couldn't perform two-factor
authentication, or forgot their password and couldn't access their account
<CONDITION> = the customer needed help with the detailed specifications
of a certain item sold on the website, such as inquiries about detailed
information of a specific retail item about materials, whether it shrinks,
stock availability, etc., but NOT inquiries about promotions, order status,
shipping status, questions about how to use the website to purchase an item,
or difficulties on using the website to add to cart or purchase an item, or
inquiries about subscription
```

**User prompt**

```
Below are the dialogs:
**dialog_0**:
[agent]: Hello! Welcome to AcmeBrands, how cani help you?
[customer]: Hi. I am very frustrated because I am trying to use your
    ↪ website and it is running SO slowly!
...
**dialog_1**:
[customer]: hi there
[agent]: Hi! What can I help you with today?
[customer]: i wanted to know if you'd be able to tell me the arm length on
    ↪ a shirt i'm thinking of buying?
...
```

**AirDialog dataset**

**System prompt** The following are dialogs between a airline ticketing agent and a customer. Dialogs start with headers such as **dialog_1**, **dialog_2**, and so on. Your task is to find out the dialogs where CONDITION. Please only output a list of indices in the format of [dialog_3, dialog_7, ...]

<CONDITION> = the agent has booked a flight for the customer (not including the flight change)
<CONDITION> = the agent canceled the existing valid reservation for the customer
<CONDITION> = the customer aims to book a flight from departure to destination but finds no flights between departure and destination
<CONDITION> = the customer wants to change or cancel the flight but there is no valid reservation under this customer

**User prompt**

```
Below are the dialogs:
**Dialog 0**:
customer: Hello.
agent: Hello, how can I help you today?
customer: Can you please find a flight from DFW to SEA?
...
**Dialog 1**:
customer: Hi, I am Melissa Thompson.
agent: Hello, how may I support you today?
customer: I want to celebrate Thanks giving day 11/23 with my friends
at New York. Can you book a ticket for me?
...
```

**Clevr dataset**

**System prompt**

```
Please read and parse the following images. Images start with labels such
as **image_0**, **image_1**, and so on.
Your task is to find out the images where <CONDITION>. Please only output
a list of indices in the format of [image_3, image_7, ...]
```

<CONDITION> = there are less than 4 objects in the image
<CONDITION> = there are more than 3 spheres in the image

**User prompt**

```
image_0: <base64_encoded_image>
image_1: <base64_encoded_image>
...
Given above, the relevant images are:
```

### D.1.3   Task: Abstraction and aggregation

**ABCD dataset**

**System prompt**

```
The following are dialogs between a customer service agent and a customer.
Dialogs start with headers such as **dialog_1**, **dialog_2**, and so on.
Your task is to analyze all the dialogs, and summarize
"<ABSTRACT_ATTRIBUTE>" into groups. Please output the table of
your analysis, in the format of pairs of ("<ABSTRACT_ATTRIBUTE>",
number_of_dialogs belong to that). Specifically in the format as:
group 1,number_of_dialogs
group 2,number_of_dialogs
...
```

```
<ABSTRACT_ATTRIBUTE> = the type of account access issue <ABSTRACT_ATTRIBUTE>
= the single item involved in the dialog
```

**User prompt**

```
Below are the dialogs:
**dialog_0**:
[agent]: good afternoon, how can I help you?
[customer]: hey think i mixed up or forgot which username I'm in with you
    ↪ guys as
...
**dialog_1**:
[agent]: Hi there, thanks for contacting Acme! How can I help you?
...
```

**AirDialog dataset**

**System prompt**

```
The following are dialogs between a airline ticketing agent and a customer.
Dialogs start with headers such as **dialog_1**, **dialog_2**, and so on.
Your task is to analyze all the dialogs, and summarize
"<ABSTRACT_ATTRIBUTE>" into groups.  Please output the table of
your analysis, in the format of pairs of ("<ABSTRACT_ATTRIBUTE>",
number_of_dialogs belong to that).  Specifically in the format as:
group 1,number_of_dialogs
group 2,number_of_dialogs
...

<ABSTRACT_ATTRIBUTE> = the outcome of the dialog
```

**User prompt**

```
Below are the dialogs:
**dialog_0**:
customer: Hi.
agent: Hello. How may I help you?
customer: I need to book a flight ticket from DEN to EWR to enjoy music
    ↪ festivals.
...
**dialog_1**:
customer: Hi.
agent: Hello, how may I help you?
...
```

## D.2 Prompts for UQE-orchestrated LLMs

### D.2.1 Task: Conditional aggregation, Semantic retrieval

**IMDB dataset**

**System prompt**

```
Please analyze the following movie review, and only reply <True> if
<WHERE_CLAUSE>, or <False> otherwise.

<WHERE_CLAUSE> = the review sentiment is overall positive
```

**User prompt**

```
[Movie review]:  May I please have my $13.00 back?  I would have rather
watched "Hydro- Electric Power Comes to North America"...
```

**ABCD dataset**

**System prompt**

Read the following customer support dialog between an agent and a customer, and only reply <True> if <WHERE_CLAUSE>, or <False> otherwise.

<WHERE_CLAUSE> = the customer could not access the account and was locked out, such as couldnt́ recall their username, couldn't perform two-factor authentication, or forgot their password and couldn't access their account
<WHERE_CLAUSE> = the customer needed help with the detailed specifications of a certain item sold on the website, such as inquiries about detailed information of a specific retail item about materials, whether it shrinks, stock availability, etc., but NOT inquiries about promotions, order status, shipping status, questions about how to use the website to purchase an item, or difficulties on using the website to add to cart or purchase an item, or inquiries about subscription

**User prompt**

```
[Dialog]:  [agent]:  Hi!  Thank you for contacting us today.  How can I
help you?
[customer]:  I'm pretty upset that a jacket that I ordered is now saying
that it is out of stock.  Do you know when it will be back in stock?
[agent]:  I am so sorry that happened to you
[agent]:  Yes, let me look in to that for you
[action]:  Searching the FAQ pages ...
...
```

**AirDialog dataset**

**System prompt**

Read the following airline ticketing dialog between the customer and the agent, and only reply <True> if <WHERE_CLAUSE>, or <False> otherwise.

<WHERE_CLAUSE> = the agent has booked a flight for the customer (not including the flight change)
<WHERE_CLAUSE> = the agent canceled the existing valid reservation for the customer
<WHERE_CLAUSE> = the customer aims to book a flight from departure to destination but finds no flights between departure and destination
<WHERE_CLAUSE> = the customer wants to change or cancel the flight but there is no valid reservation under this customer

**User prompt**

```
[Dialog]:  customer:  Hello.
agent:  Hello, how can I help you?
customer:  Please book a flight ticket from CLT to DEN.
agent:  Sure, let me know your travelling dates.
...
```

**Clevr dataset**

**System prompt**

Read the following image, and only reply <True> if <WHERE_CLAUSE>, or <False> otherwise.

<WHERE_CLAUSE> = there are less than 4 objects in the image
<WHERE_CLAUSE> = there are more than 3 spheres in the image

**User prompt**

```
Image:  <base64_encoded_image>
```

### D.2.2   Task: Abstraction and aggregation

**ABCD dataset**

    1. **Building taxonomy**

**System prompt**

```
The following are dialogs between a customer service agent and
a customer.  Dialogs start with headers such as **dialog_1**,
**dialog_2**, and so on.
Your task is to analyze all the dialogs, and summarize
"<ABSTRACT_ATTRIBUTE>" into groups.  Please output the table of
your analysis, in the format of pairs of ("<ABSTRACT_ATTRIBUTE>",
number_of_dialogs belong to that).  Specifically in the format as:
group 1,number_of_dialogs
group 2,number_of_dialogs
...
<ABSTRACT_ATTRIBUTE> = the type of account access issue
<ABSTRACT_ATTRIBUTE> = the single item involved in the dialog
```

**User prompt**

```
    Below are the dialogs:
    **dialog_0**:
    [agent]: good afternoon, how can I help you?
    [customer]: hey think i mixed up or forgot which username I'm in
        ↪ with you guys as
    ...
    **dialog_1**:
    [agent]: Hi there, thanks for contacting Acme! How can I help you
        ↪ ?
    ...
```

2. **Group-wise conditional aggregation**

**System prompt**

```
    Read the given airline ticketing dialog between an agent and a
        ↪ customer, and classify issue, into one or several
        ↪ categories below. Here are the description of the 2
        ↪ categories:
    [0]: Forgot Username
    [1]: Forgot PasswordOnly reply the index of the category,
        ↪ separated by ",". Here is the example format:
    [0, 3]
```

**User prompt**

```
    Here is the customer support dialog:
    [agent]: Hello, how can i help you today
    [customer]: Hi. I seem to have forgotten my username
    [agent]: Okay lets get that for you, could i get your Full Name
        ↪ Zip Code Email Adress and Phone Number please
    [customer]: Sanya Afzal
```

**AirDialog dataset**

1. **Building taxonomy**

**System prompt**

```
The following are dialogs between a airline ticketing agent and
a customer.  Dialogs start with headers such as **dialog_1**,
**dialog_2**, and so on.
Your task is to analyze all the dialogs, and summarize
"<ABSTRACT_ATTRIBUTE>" into groups.  Please output the table of
your analysis, in the format of pairs of ("<ABSTRACT_ATTRIBUTE>",
number_of_dialogs belong to that).  Specifically in the format as:
group 1,number_of_dialogs
```

```
group 2,number_of_dialogs
...
```
<ABSTRACT_ATTRIBUTE> = the outcome of the dialog

**User prompt**

```
Below are the dialogs:
**dialog_0**:
customer: Hi.
agent: Hello. How may I help you?
customer: I need to book a flight ticket from DEN to EWR to enjoy
    ↪  music festivals.
...
**dialog_1**:
customer: Hi.
agent: Hello, how may I help you?
...
```

2. **Group-wise conditional aggregation**

**System prompt**

```
Read the given airline ticketing dialog between the customer and
    ↪ the agent, and classify outcome, into one or several
    ↪ categories below. Here are the description of the 2
    ↪ categories:
[0]: Reservation cancelled
[1]: Ticket bookedOnly reply the index of the category, separated
    ↪  by ",". Here is the example format:
[0, 3]
```

**User prompt**

```
Here is the airline ticketing dialog: customer: Hello
agent: Hello, how may I help you?
customer: Can you help me to book a flight ticket from SEA to AUS
    ↪ ?
agent: Sure, we are glad to help you. May I know your travelling
    ↪ dates?
```

