# OpenReview forum: "UQE: A Query Engine for Unstructured Databases"
_NeurIPS.cc/2024/Conference — NeurIPS 2024 poster_

### Official Review · Reviewer_E5QU · 2024-07-11

**Soundness:** 3
**Presentation:** 4
**Contribution:** 2
**Rating:** 5
**Confidence:** 4

**Summary:**

This paper proposed a Universal Query Engine to directly draw insights from unstructured data. For aggregation queries, this paper designed an unbiased sampling algorithm to solve the problem that it’s hard to apply index on virtual columns. Inspired by the workflow of the C++ compiler, this paper also proposed a workflow of optimizing and generating prompt to achieve more efficient query.

**Strengths:**

1. This paper suited an interesting and important problem, by using LLM, more complex, semantics -related SQL queries can be achieved.
2. This paper took efficient into consideration and proposed corresponding algorithm and workflow to improve the query speed.
3. Provided theory proof of the unbiased sampling algorithm.
4. The idea of introducing compiler’s workflow into such problem to accelerating the LLM’s inference is interesting.

**Weaknesses:**

1. UQE seems to lack scalability on the Semantic Retrieval task. The latency reported in Table 5 is as high as 38.08 seconds. Also, the paper didn’t state that whether these numbers are the total latency or average latency per query. If they are latency per query, using UQE on the Semantic Retrieval task still has a far distance from practice.
2. The cost of embedding operation of Algorithm 1 and 2, and the cost of Algorithm 2 are not discussed in the paper. Although the authors wrote in the appendix that the g function in Algorithm 2 caused the high latency problem, no quantitative analysis was provided.
3. Lack of comparison with important baselines. I noticed that the references [9] and [33] mentioned in related work are not included in the baselines. Using LLM as UDF may significantly slow down the queries, however, they may have better accuracy on aggregation queries. Thus, I think it’s necessary to include them in the baselines to further evaluate the performance of the trade-off made by Algorithm 1 between accuracy and latency.
4. Lack of evaluation of UQE’s time cost in the main text of the paper. I noticed those results are reported in the appendix. However, in the database area, the latency and cost are significant metrics that should be reported in the main text of the paper.

**Questions:**

I noticed that lc-gpt-4-turbo also has quite a high latency on Semantic Retrieval tasks, is it possible to do more optimization on the model itself to improve the performance to make such a method practice?

**Limitations:**

Yes, the authors have mentioned the limitations of this work, but I suggest to provide more comprehensive evaluations, detailed discussions about UQE’s scalability and cost. After all, these are critical problems about the practicality.

---

> ### Author Rebuttal · Authors · 2024-08-07
>
> We thank the reviewer for the insightful comments, and we do value your opinion regarding the latency issues. Below we try to justify from the use cases and existing baselines, and also provide more experimental results.
>
> ### Q1 **"...latency...far distance from practice..."**
>
> The latency is per query and is averaged over 8 runs. We agree that this latency is not meant for real-time queries like the SQL engine behind web apps. The type of query that is most useful is for massive data analytics, where the alternative solution is usually to train a task specific ml model, build up the data preprocessing pipeline and then run SQL, which can easily take days for data scientists.
>
>
> Secondly, our framework offers the flexibility to do trade-offs between latency and accuracy. Take the IMDB retrieval experiment as an example:
>
>
> | Budget $B$ | 256           | 128           | 64            | 32            |
> | ---------- | ------------- | ------------- | ------------- | ------------- |
> | Latency(s)   | 38.08         | 21.61         | 11.11         | 5.84          |
> | F1 score   | 0.978 ± 0.003 | 0.974 ± 0.005 | 0.921 ± 0.013 | 0.828 ± 0.035 |
>
> And for aggregation:
>
> | Budget $B$ | 128            | 64            | 32            |
> | ---------- | -------------- | ------------- | ------------- |
> | Latency(s) |  5.83 |  4.28      | 2.93            |
> | Error      | 5.75% ± 3.43%  | 6.84% ± 5.52% | 8.29% ± 7.39% |
>
> So comparing to alternative solutions for data analytics, which take days to first preprocess the data into structured columns and then run SQL, the speed of UQE is significantly improved.
>
> ### Q2 **"...cost of embedding... cost of Algorithm 2...g function..."**
>
> The embedding cost is the same as the vector DBs. This is a pre-processing step and only needs to be done once per dataset. For text we use voyage-2 which costs 0.1 USD / million tokens, and it roughly take 8h (due to rate limit) to embed these datasets. For image we use Cloud Vertex multi-modal embedding and it takes 12h (due to rate limit).
>
> For a query on IMDB that takes 21s, querying g and updating g takes 1.5s and 0.5s respectively. The main reason for the latency is the sequential nature of online learning of g, and a potential improvement is to leverage batched online learning to improve the parallelism.
>
> ### Q3 **"...baselines.. the trade-off made by Algorithm 1 between accuracy and latency"**
>
> We followed the reviewer's advice to include BINDER (the viper-gpt is similar but for images) in all the experiments, and as shown in our global rebuttal, the performance of BINDER is much worse when given the same LLM budgets (we stop the BINDER execution when running out of budget, as looping over the entire database can easily cost hundreds of dollars, which is not feasible for us).
>
> To see how the accuracy evolves for both UQE and BINDER, we perform the following experiments on aggregation tasks (IMDB) by varying the budgets. In paper we use B=128, and we lower/higher the budget for UQE/BINDER respectively in the table below.
>
>
> | Budget $B$ | 512            | 256           | 128            | 64            | 32            |
> | ---------- | -------------- | ------------- | -------------- | ------------- | ------------- |
> | BINDER     | 8.11%  ± 3.15% | 8.35% ± 5.47% | 13.67% ± 6.24% |      -         |      -         |
> | UQE        |         -       |     -          | 5.75% ± 3.43%  | 6.84% ± 5.52% | 8.29% ± 7.39% |
>
>
> we can see that both BINDER and UQE will have more accurate estimation when given more budget. But for BINDER, it takes 16x more LLM calls (512) compared to UQE (32) to obtain the same level of accuracy or precision. We hope this answers the question of tradeoff of Algorithm 1.
>
>
> ### Q4 **"...latency and cost...in the main text..."**
>
> We have provided the cost per query in every table in the main text. We will follow the reviewer's advice to include the latency and also the above trade-off analysis into the main text.
>
>
> ### Q5 **"... lc-gpt-4-turbo also has quite a high latency..."**
>
>
> lc-gpt-4-turbo is basically the baseline on feeding the entire content into gpt-4-turbo and ask questions. There could be several ways that make it faster (though it is not the focus of this paper to make LLM itself run faster):
>
> 1. prefix caching; if the data never changes, one can cache majority of the kv states of prompts which can save 2x of the cost/latency or so.
>
> 2. in general, the models will only get faster/cheaper, e.g., gpt-4o-mini which came out recently.

---

> ### Author Response · Authors · 2024-08-11
> **We'd love to hear your opinion on the rebuttal**
>
> Dear reviewer E5QU,
>
> We have provided additional experiments and explanations to address your valuable feedback.
>
> Could you please kindly take a look, and let us know if you have further concerns so that we can provide further clarifications if needed?
>
> Thank you so much!
>
> Best,

---

### Official Review · Reviewer_rs84 · 2024-07-12

**Soundness:** 2
**Presentation:** 2
**Contribution:** 2
**Rating:** 5
**Confidence:** 4

**Summary:**

This paper propose a new Universal Query Engine (UQE) that directly interrogates and draws insights from unstructured data collections.
Further, a Universal Query Language (UQL) is proposed as a dialect of SQL that provides full natural language flexibility in specifying conditions and operators.
UQE leverages the ability of LLMs to conduct analysis of unstructured data and achieve efficient and accurate query execution.

**Strengths:**

1. The idea of designing new query language and query engine for unstructured databases leveraging the power of LLMs is interesting.
2. The paper is also very clear with thorough experiments and analysis.
3. Well-organized presentation of the proposed method and its components.

**Weaknesses:**

1. Many details about the design and implementation of UQE are missing. For example, the description of the generation of machine-specific code (assembly instructions) during the compilation stage (L251) is not clear. Please clarify the details of this strategy. Additionally, the authors claim that calling the LLM dominates the overall cost per query (L239). Providing detailed information about the LLM-calling stage would be beneficial.
2. The indexing technique introduced by UQE is related to the sampling strategy based on specific queries. However, database indexing is traditionally meant to facilitate quick searches. If each query requires sampling, will this impact efficiency? Additionally, doesn't this compromise accuracy?
3. This paper primarily addresses unstructured datasets, including video and audio. However, the experiments only involve text and images, lacking validation for broader generalizability.
4. Furthermore, there are plenty of multi-modal vector databases, yet the authors only compare the RAG-based query. Can UQE outperform recent multi-modal query databases, such as Milvus and MillenniumDB? If so, demonstrating this would make the evaluation more convincing.
5. Does the query support join operations, such as joining across multiple different modal files?
6. It's hard to validate the results without supplying code/models.

**Questions:**

Please see the weakness section.

**Limitations:**

No discussion on limitations, I suggest the authors to add one.

---

> ### Author Rebuttal · Authors · 2024-08-07
>
> We thank the reviewer for the valuable comments. Below we try to address the potential misunderstandings, and provide more experimental results for justifications.
>
>
> ### Q1 **"...machine-specific code...clarify the details..."**
>
> The analogy is to show how to convert the query (in UQL) to the machine code (the concrete prompts that work with LLMs). Concretely, for example, a `WHERE cond` statement would be "compiled" to
>
> >*Please analyze the following movie review, and only reply <True> if cond, or <False> otherwise.*
>
> and executed on the rows sampled by the online-learning or unbiased samplers. We provided the full set of prompts in appendix for your information.
>
> ### Q1 **"...LLM dominates the overall cost per query...Providing detailed information"**
>
> For example, a query on clevr dataset costs  0.2USD using gpt-4o. The compute on the client side only takes CPU time for 30s. On AWS such a machine costs 5.80USD/month, so 30s of compute would be nothing, compared to the LLM calls.
>
> ### Q2 **"...database indexing...sampling...impact efficiency? ...accuracy?"**
>
> The traditional database is able to build index on the columns that might be conditioned on, so as to avoid linear scan of the entire database. For unstructured data, as we don't know the query beforehand, there is no such indexing. Vector DB builds the embedding based indexing, but can only be useful for limited types of queries as we have seen in the experiments.
>
> Based on that, we propose the analogy of indexing in unstructured database, i.e. the sampling techniques to 1) avoid linear scan; 2) handle complex reasonings; 3) have statistical guarantee.
>
> So to answer your question, it is actually the sampling (and the online learning) that makes things efficient. There's rich literature [1] on approximated compute engine, and we have shown that its accuracy is much better than alternatives throughout comprehensive experiments (e.g., **40x** F1 score compared to the reviewer suggested Milvus for some queries).
>
>
> ### Q3 **"...addresses unstructured datasets, including video and audio. However, the experiments only involve text and images"**
>
> The proposed UQE framework rely on the LLM backend for multi-modal queries, and the optimization done in the paper is agnostic to the modality.
>
> Nevertheless, we provide new experimental results on the Audio MNIST, which contains 30k wav files from 60 speakers pronouncing digits 0-9.
>
> Below we perform the audio semantic retrieval experiments. The query is first converted to audio space using TTS and the corresponding audio embedding is used for MIPS/Milvus search.
> | Query                                    | BINDER-gemini | MIPS/Milvus | UQE-gemini    |
> | ---------------------------------------- | ------------- | ----------- | ------------- |
> | "Three"                                  | 0.109 ± 0.022 | 0.445       | **0.839 ± 0.135** |
> | "A number whose square equals to itself" | 0.259 ± 0.068 |    0.039         | **0.922 ± 0.024**             |
> | "The minimum prime number"               | 0.107 ± 0.044 | 0           | **0.917 ± 0.025** |
>
> We can see UQE consistently does better than alternatives, and it also allow complex queries that require reasoning, while embedding based MIPS or Milvus is limited to the type of queries.
>
> ### Q4 **"...Can UQE outperform ... Milvus and MillenniumDB..."**
>
> In our paper we compared against MIPS (Maximum inner-product search) and we show the clear advantage in Table 2. Vector DB like Milvus is designed for fast but approximated nearest neighbor (ANN) search. By default it uses inner product as the similarity metric, thus is expected to be an approximation of MIPS in our paper (where we do full vector database scan without approximation).
>
> Nevertheless, we still run Milvus on all benchmarks, and the results have been included in the global response and UQE still wins in almost all query types. MillenniumDB is for graph or structured data query and thus may not be suitable to compare. We hope this would resolve some misunderstandings.
>
>
> ### Q5 **"...join operations..."**
>
> No, we have stated in our conclusion section that table join optimization is not done in our paper. With that said, one can still create the virtual columns in UQL via `SELECT` and then use existing SQL engine for the joins, at a potential very high cost of language model calls. We hope to optimize the join operation in future work.
>
> ### Q6 **"...code/models"**
>
> The model/data and prompts are all provided already. We will release the code.
>
> ### **Limitation section**
>
> We have already clearly stated the limitation of current work in the conclusion section, including 1) lack of table join optimization; 2) LLM selection; 3) even larger datasets. And we will work on these in future works.
>
>
> References:
> >[1] A handbook for building an approximate query engine.

---

> ### Author Response · Authors · 2024-08-11
> **We'd love to hear your opinion on the rebuttal**
>
> Dear reviewer rs84,
>
> We have provided additional experiments and explanations to address your valuable feedback.
>
> Could you please kindly take a look, and let us know if you have further concerns so that we can provide further clarifications if needed?
>
> Thank you so much!
>
> Best,

---

> > ### Comment · Reviewer_rs84 · 2024-08-14
> >
> > Thanks for your quick response! Most of my concerns are resolved now. I am increasing my score to 5.

---

### Official Review · Reviewer_6SvA · 2024-07-13

**Soundness:** 4
**Presentation:** 4
**Contribution:** 3
**Rating:** 7
**Confidence:** 3

**Summary:**

This paper proposes an ambitious new framework for analytics on unstructured databases, the Universal Query Engine (UQE). The authors first present a list of semantics and clauses for querying unstructured databases and propose methods for implementing these functionalities, including indexing and compiling. Experiments on multimodal unstructured data analytics tasks show that UQE outperforms other methods, demonstrating the promise of utilizing UQE in database analytics.

**Strengths:**

- The motivation is quite interesting and necessary to broaden the functionalities of databases.
- The paper is clearly written and well-presented.
- The authors presented the exact scope of the task and proposed methods accordingly.
- The proposed approach is reasonable and performs well within the scope of the work.

**Weaknesses:**

- Could you elaborate more on the difference between BINDER and this approach (line 39)? Is it due to the lack of indexing and compatibility with unstructured information?
- Similar to the first point, the baselines seem a bit weak. Why have the authors not used BINDER as their baseline?
- I want to hear the authors' opinions on how advancements in LLMs and vision models might affect the guarantee that databases consistently return the same results. Traditional database engines, which are widely used, provide this consistency over time. Could these technological advancements affect the reliability of database analytics and, subsequently, compromise their reliability?

**Questions:**

I have listed my questions in the Weaknesses section.

**Limitations:**

The limitations of the work are properly stated.

---

> ### Author Rebuttal · Authors · 2024-08-06
>
> We thank the reviewer for providing insightful and constructive feedback. We provide our response below, as well as extra experimental results in the global response. We look forward to learning your further thoughts.
>
> ### **"... difference between BINDER..."**
>
> Yes as the reviewer pointed out, our focus is on the analogy of "indexing" (or sampling and search) to efficiently implement the unstructured database query engine. BINDER focuses on translating natural language questions into programs with llm as a callable function. The execution of this program is delegated to existing SQL implementations. These two can actually work together to deliver a better end to end system.
>
> ### **"...used BINDER as their baseline?"**
>
> While BINDER and our UQE have different focuses, we provided additional experiments using BINDER on our tasks in the general response. We can see that without the query engine optimization, BINDER would achieve much higher variance and error when performing aggregation queries, and the retrieval F1 is also much lower compared to UQE under the same LLM budgets.
>
> ### **"...consistently return the same results...compromise their reliability"***
>
> We totally agree with the reviewer that the consistency and reliability is important, and that's why our algorithm (especially the one for the aggregation query) optimizes for low-variance and zero-bias estimation, which has significantly boosted this compared to all baselines. In addition to that, one can further reduce the temperature of the LLMs and use fixed random seeds to make sure the results are reproducible for a given setup.
>
> The reviewer raised another good point where the LLM behind the scene gets updated and that would cause the inconsistency across different versions of LLMs. This is a generic problem of the application built on top of LLMs, but here are several ways to mitigate it.
>
> 1. Result caching: for the same query, one can always cache the intermediate or final results to save the compute and also make it consistent for the same question.
>
> 2. Prompt calibration: once a new model is deployed, a common practice is to adjust the prompt a bit to make it consistent or fitting better on the target tasks. Similar practices can be made for the database queries, as long as a proper evaluation set is available. In our experiment we prepare multi-modal datasets with the ground-truth labels to validate and calibrate the prompts.
>
> 3. Even without any further effort, it may not be that inconsistent in some situations. In our paper we used claude-3-haiku as the backbone, and below we use the latest gpt-4o-mini to provide more justification for that. On IMDB we see little variation, where we can see that, the difference caused by the model switching is much lower than using a worse query engine.
>
>
> | Task        | BINDER         | UQE-haiku (in paper) | UQE-gpt4o-mini (new) |
> | ----------- | -------------- | -------------------- | -------------------- |
> | Retrieval   | 0.505 ± 0.030  | 0.978 ± 0.003        | 0.956 ±0.010         |
> | Aggregation | 13.67% ± 6.24% | 5.75% ± 3.43%        | 6.33%± 4.71%         |
>
> Of course, this behavior shift would be task related, but with the advances of LLMs we believe this variation would converge and be more stable to prompts.

---

> > ### Comment · Reviewer_6SvA · 2024-08-09
> >
> > Thanks for the response. It cleared up my concerns.

---

> > > ### Author Response · Authors · 2024-08-12
> > > **thank you!**
> > >
> > > Thank you reviewer 6SvA for your kind reply and your recognition of the paper!

---

### Official Review · Reviewer_uiEW · 2024-07-18

**Soundness:** 2
**Presentation:** 2
**Contribution:** 2
**Rating:** 6
**Confidence:** 2

**Summary:**

This paper proposes an unstructured query language based on a small segment of SQL. Its key feature is that it further supports unstructured texts and images because the authors assume that LLMs work on intra-row texts. During the query, the authors propose using online learning on LLM outputs over batches to improve the sample quality in the upcoming batches. Empirical results convey important information; that is, feeding longer context into larger LLMs does not work as well as using smaller LLMs in a more organized way.

**Strengths:**

This paper discusses several possibilities for using LLMs in relational databases with text columns. It reveals a new possibility that one could do something between 1. using LLMs as a black box and letting them handle everything in the context. and 2. using LLMs to parse the question into SQL and let the database engine do the job with much smaller LLMs.

For queries requiring sampling, a simple online learning algorithm is used to improve the sample efficiency.

**Weaknesses:**

- This paper is more like a technical report or a guideline for building certain products. The overall product presented is simplified from the introduction's proposed question. Realizing this system under the strong assumption that LLMs work on intra-row semantic understanding is not too difficult: the virtual column, although it seems to be filled by LLMs, can be realized in several prompts, such as closed-domain classification for WHERE and open-domain classification for GROUPBY.

**Questions:**

- Given that the output in the virtual column will work with other parts of the SQL, what is the failure rate in output formatting in the experiments?

**Limitations:**

Yes,

---

> ### Author Rebuttal · Authors · 2024-08-06
>
> We thank the reviewer for the valuable comments. Below we try to address the potential misunderstandings, and provide more experimental results for justifications.
>
>
> ### **"...technical report...the strong assumption that LLMs work on intra-row semantic understanding"**
>
> While generating the virtual column entirely and process with SQL is a feasible approach, the goal of the paper is to present an efficient system that can void such kind of "linear scan" of the database content, and thus the core of the contribution lies in the unbiased+low-variance sampler and the online learning approach that orchestrates well with the LLMs. Thus we focus on providing algorithmic detail and the proof (see appendix A) for these while put less attention on the database and system layer. If the reviewer have specific details that would like to learn more, please feel free to let us know. We will also release the code afterwards.
>
> Regarding the assumption that LLMs work on intra-row semantic understanding, we have carefully examined this. On IMDB dataset, the gpt model can achieve **97%** accuracy on average over each review, and many of the errors are actually due to the unclear sentiment from the data. Definitely the accuracy would depend on the difficulty of the question, but having a good-enough out-of-the-box solution can be a good starting point, given that people are still pushing the frontier of LLMs.
>
>
> ### **"...failure rate in output formatting..."**
>
> We completely rewrite the entire SQL engine and thus the error handling can be easily done in our system. As is shown in Appendix, most of the time we ask LLMs to output <True> or <False> based on the query. We did a quick experiment to see when the LLM failed to generate either of these.
>
>
> | Dataset | LLM            | Violation rate |
> | ------- | -------------- | -------------- |
> | IMDB    | Claude-3-haiku | 0.0%           |
> |         | gpt-4o-mini    | 0.8%           |
> | ABCD    | Claude-3-haiku | 0.0%           |
> |         | gpt-4o-mini    | 0.2%           |
>
> We can see the above models, despite being small and cheap, are very good at instruction following and the formatting violation rate can be tolerated with extra error handling. And we believe that the next generation LLMs can only be better at instruction following. Actually as of Aug-6, openai released a new version of gpt-4o (**gpt-4o-2024-08-06**) which claimed to have **100%** accuracy on complex format following.

---

> ### Author Response · Authors · 2024-08-11
> **We'd love to hear your opinion on the rebuttal**
>
> Dear reviewer uiEW,
>
> We have provided additional experiments and explanations to address your valuable feedback.
>
> Could you please kindly take a look, and let us know if you have further concerns so that we can provide further clarifications if needed?
>
> Thank you so much!
>
> Best,

---

### Author Rebuttal · Authors · 2024-08-06

We thank all the reviewers for their insightful feedback, and below we provide experimental updates w.r.t baselines to address the reviewers’ comments.

### **Comparison with baselines like BINDER [1] (review 6SvA and E5QU)**

While [1] and UQE use similar SQL-like language, the focus is very different. [1] or [2] focuses on translating natural language questions into programs with llm as a callable function. The execution of this program is delegated to existing SQL implementations. On the contrary, UQE replaces the query engine layer, with the focus to execute this kind of programs in an efficient and accurate manner.

Nevertheless, we adapted BINDER (or equivalently [2] if it is for image) for our task by providing the program directly and executing it until it reaches the same API cost as UQE.

Below is the estimation error and variance on aggregation tasks (lower the better).

| Benchmark |       Query        |  lc-GPT4-turbo  |     BINDER      |         UQE         |
|:---------:|:------------------:|:---------------:|:---------------:|:-------------------:|
|   IMDB    | sentiment_positive | 49.02% ± 21.23% | 13.67% ± 6.24%  |  **5.75% ± 3.43%**  |
|   ABCD    |   account_access   | 69.25% ± 32.82% | 18.99% ± 9.85%  | **11.75% ± 9.78%**  |
|           | single_item_query  | 78.42% ± 9.36%  | 26.95% ± 22.16% | **12.32% ± 10.53%** |
| AirDialog |        book        | 47.58% ± 15.24% | 10.15% ± 7.64%  |  **4.98% ± 2.26%**  |
|           |     no_flight      | 47.92% ± 21.62% | 21.08% ± 16.78% |  **8.78% ± 8.12%**  |
|           |   no_reservation   | 50.54% ± 21.86% | 21.19% ± 12.10% |  **7.23% ± 5.40%**  |
|   Clevr   |   obj_count < 4    | 22.46% ± 19.35% | 31.04% ± 25.15% |  **9.55 ± 8.55%**   |
|           |   # spheres > 3    | 35.72% ± 14.95% | 19.35% ± 13.81% | **15.14% ± 10.71%** |

Below is the F1 score (higher the better) on semantic retrieval tasks. Note that vector DB (or per reviewer rs84, Milvus) is doing approximated MIPS as we reported in the paper, so basically they are the same and we merge their results below:

| Benchmark |       Query        | lc-GPT4-turbo  |     BINDER     | MIPS/Milvus |        UQE        |
|:---------:|:------------------:|:--------------:|:--------------:|:-----------:|:-----------------:|
|   IMDB    | sentiment_positive | 0.397 ± 0.041  | 0.505 ± 0.030  |    0.875    | **0.978 ± 0.003** |
|   ABCD    |   account_access   | 0.045 ± 0.033  | 0.076 ± 0.017  |  **0.961**  |   0.940 ± 0.019   |
|           | single_item_query  | 0.023 ± 0.021  | 0.065 ± 0.017  |    0.266    | **0.935 ± 0.006** |
| AirDialog |        book        | 0.327 ± 0.0667 | 0.342 ±  0.031 |    0.930    | **0.979 ± 0.010** |
|           |     no_flight      | 0.066 ± 0.037  | 0.144 ±  0.034 |    0.867    | **0.928 ± 0.018** |
|           |   no_reservation   | 0.156 ± 0.075  | 0.145 ± 0.042  |    0.965    | **0.969 ± 0.004** |
|           |       cancel       | 0.006 ± 0.009  | 0.013 ±  0.009 |    0.066    | **0.741 ± 0.205** |
|   Clevr   |   obj_count < 4    | 0.058 ± 0.026  | 0.093 ± 0.031  |    0.023    | **0.897 ± 0.006** |
|           |   # spheres > 3    | 0.037 ± 0.027  | 0.089 ± 0.017  |    0.145    | **0.859 ± 0.007** |

From the tables above, we can see UQE achieves much better estimation precision and retrieval effectiveness compared to BINDER or a straight query engine implementation. Meanwhile, we confirmed again that vector DBs, even the commercial ones, may not be suitable for some retrieval tasks that require deep reasoning.

To see how the accuracy evolves for both UQE and BINDER, we perform the following experiments on aggregation tasks (IMDB) by varying the budgets. In paper we use B=128, and we lower/higher the budget for UQE/BINDER respectively in the table below.


| Budget $B$ | 512            | 256           | 128            | 64            | 32            |
| ---------- | -------------- | ------------- | -------------- | ------------- | ------------- |
| BINDER     | 8.11%  ± 3.15% | 8.35% ± 5.47% | 13.67% ± 6.24% |      -         |      -         |
| UQE        |         -       |     -          | 5.75% ± 3.43%  | 6.84% ± 5.52% | 8.29% ± 7.39% |

we can see that both BINDER and UQE will have more accurate estimation when given more budget. But for BINDER, it takes 16x more LLM calls (512) compared to UQE (32) to obtain the same level of accuracy or precision.

### **Results on audio database (review rs84)**

Follow reviewer rs84, we provide new experimental results on the Audio MNIST, which contains 30k wav files from 60 speakers pronouncing digits 0-9.

Below we perform the audio semantic retrieval experiments. The query is first converted to audio space using TTS and the corresponding audio embedding is used for MIPS/Milvus search.
| Query                                    | BINDER-gemini | MIPS/Milvus | UQE-gemini    |
| ---------------------------------------- | ------------- | ----------- | ------------- |
| "Three"                                  | 0.109 ± 0.022 | 0.445       | **0.839 ± 0.135** |
| "A number whose square equals to itself" | 0.259 ± 0.068 |    0.039         | **0.922 ± 0.024**             |
| "The minimum prime number"               | 0.107 ± 0.044 | 0           | **0.917 ± 0.025** |

We can see UQE consistently does better than alternatives, and it also allow complex queries that require reasoning, while embedding based MIPS or Milvus is limited to the type of queries.


>References\
[1] Binder: Binding Language Models in Symbolic Languages\
[2] ViperGPT: Visual Inference via Python Execution for Reasoning

---

### Author Response · Authors · 2024-08-14
**last reminder on adding your thoughts on NeurIPS authors' feedback**

Dear reviewers,

We sincerely appreciate your effort during the review process. As the deadline for reviewer-author discussion is approaching in several hours, we would like to send out a last reminder to see if you have any quick follow up questions to our rebuttal, so that we might still be able to respond.

Thank you so much for your understanding!

Best,
Authors

---

### Decision · Program_Chairs · 2024-09-25

**Decision:**

Accept (poster)

**Comment:**

The paper presents a system that can be used to conduct data analytics on unstructured data. The system uses, of course, LLMs :) The system is tested and thoroughly assessed.

Reviewers have given average scores but in general we are all favorable for this paper to appear as poster at NeurIPS.